# Coresets for Clustering with Missing Values

**Vladimir Braverman**
Johns Hopkins University
vova@cs.jhu.edu

**Shaofeng H.-C. Jiang**
Peking University
shaofeng.jiang@pku.edu.cn

**Robert Krauthgamer**
Weizmann Institute of Science
robert.krauthgamer@weizmann.ac.il

**Xuan Wu**
Johns Hopkins University
wu3412790@gmail.com

## Abstract

We provide the first coreset for clustering points in $\mathbb{R}^d$ that have multiple missing values (coordinates). Previous coreset constructions only allow one missing coordinate. The challenge in this setting is that objective functions, like $k$-MEANS, are evaluated only on the set of available (non-missing) coordinates, which varies across points. Recall that an $\epsilon$-coreset of a large dataset is a small proxy, usually a reweighted subset of points, that $(1 + \epsilon)$-approximates the clustering objective for every possible center set.

Our coresets for $k$-MEANS and $k$-MEDIAN clustering have size $(jk)^{O(\min(j,k))}(\epsilon^{-1}d\log n)^2$, where $n$ is the number of data points, $d$ is the dimension and $j$ is the maximum number of missing coordinates for each data point. We further design an algorithm to construct these coresets in near-linear time, and consequently improve a recent quadratic-time PTAS for $k$-MEANS with missing values [Eiben et al., SODA 2021] to near-linear time.

We validate our coreset construction, which is based on importance sampling and is easy to implement, on various real data sets. Our coreset exhibits a flexible tradeoff between coreset size and accuracy, and generally outperforms the uniform-sampling baseline. Furthermore, it significantly speeds up a Lloyd's-style heuristic for $k$-MEANS with missing values.

## 1 Introduction

We consider coresets and approximation algorithms for $k$-clustering problems, particularly $k$-MEANS[1] and more generally $(k, z)$-CLUSTERING (see Definition 2.1), for points in $\mathbb{R}^d$ with *missing values (coordinates)*. The presence of missing values in data sets is a common phenomenon, and dealing with it is a fundamental challenge in data science. While data imputation is a very popular method for handling missing values, it often requires prior knowledge which might not be available, or statistical assumptions on the missing values that might be difficult to verify [All01, LR19]. In contrast, our worst-case approach does not requires any prior knowledge. Specifically, in our context of clustering, the distance $\mathrm{dist}(x, c)$ between a clustering center point $c$ and a data point $x$ is evaluated only on the available (i.e., non-missing) coordinates. Similar models that aim to minimize clustering costs using only the available coordinates have been proposed in previous work [HB01, Wag04, CCB16, WLH$^+$19], and some other relevant works were discussed in a survey [HC10].

Clustering under this distance function, which is evaluated only on the available coordinates, is a formidable computational challenge, because distances do not satisfy the triangle inequality, and

---

[1]In the usual $k$-MEANS problem (without missing coordinates), the input is a data set $X \subset \mathbb{R}^d$ and the goal is to find a center set $C \subset \mathbb{R}^d, |C| = k$ that minimizes the sum of squared distances from every $x \in X$ to $C$.

35th Conference on Neural Information Processing Systems (NeurIPS 2021).

therefore many classical and effective clustering algorithms, such as $k$-MEANS++ [AV07], cannot be readily applied or even be defined properly. Despite the algorithmic interest in clustering with missing values, the problem is still not well understood and only a few results are known. In a pioneering work, Gao, Langberg and Schulman [GLS08] initiated the algorithmic study of the $k$-CENTER problem with missing values. They took a geometric perspective and interpreted the $k$-CENTER with missing values problem as an affine-subspace clustering problem, and followup work [GLS10, LS13] has subsequently improved and generalized their algorithm. Only very recently, approximation algorithms for objectives other than $k$-CENTER, particularly $k$-MEANS, were obtained for the limited case of at most one missing coordinate in each input point [MF19] or for constant number of missing coordinates [EFG+21].

We focus on designing coresets for clustering with missing values. Roughly speaking, an $\epsilon$-coreset is a small proxy of the data set, such that the clustering objective is preserved within $(1 \pm \epsilon)$ factor for all center sets (see Definition 2.2 for formal definition). Efficient constructions of small $\epsilon$-coresets usually lead to efficient approximations schemes, since the input size is reduced to that of the coreset, see e.g. [HJLW18, FRS19, MF19]. Moreover, apart from speeding up approximation algorithms in the classical setting (offline computation), coresets can also be applied to design streaming [HM04, FS05, BFL+17], distributed [BEL13, RPS15, BLK18], and dynamic algorithms [Cha09, HK20], which are effective methods/models for dealing with big data, and recently coresets were used even in neural networks [MOB+20].

## 1.1 Our Results

**Coresets.** Our main result, stated in Theorem 1.1, is a near-linear time construction of coresets for $k$-MEANS with missing values. Here, an $\epsilon$-coreset for $k$-MEANS for a data set $X$ in $\mathbb{R}^d$ with missing coordinates is a weighted subset $S \subseteq X$ with weights $w : S \to \mathbb{R}_+$, such that

$$\forall C \subset \mathbb{R}^d, |C| = k, \qquad \sum_{x \in S} w(x) \cdot \mathrm{dist}^2(x, C) \in (1 \pm \epsilon) \sum_{x \in X} \mathrm{dist}^2(x, C),$$

where $\mathrm{dist}(x, c) := \sqrt{\sum_{i : x_i \text{ not missing}} (x_i - c_i)^2}$, and $\mathrm{dist}(x, C) := \min_{c \in C} \mathrm{dist}(x, c)$; note that the center set $C$ does not contain missing values. More generally, our coreset also works for $(k, z)$-CLUSTERING, which includes $k$-MEDIAN (see Definition 2.1 and Definition 2.2). Throughout, we use $\tilde{O}(f)$ to denote $O(f \operatorname{poly} \log f)$.

**Theorem 1.1** (Informal version of Theorem 3.1)**.** *There is an algorithm that, given $0 < \epsilon < 1/2$, integers $d, j, k \geq 1$, and a set $X \subset \mathbb{R}^d$ of $n$ points each having at most $j$ missing values, it constructs with constant probability an $\epsilon$-coreset for $k$-MEANS on $X$ of size $m = (jk)^{O(\min\{j,k\})} \cdot (\epsilon^{-1} d \log n)^2$, and runs in time $\tilde{O}\left((jk)^{O(\min\{j,k\})} \cdot nd + m\right)$.*

Our coreset size is only a low-degree polynomial of $d, \epsilon$ and $\log n$, and can thus deal with moderately-high dimension or large data set. The dependence on $k$ (number of clusters) and $j$ (maximum number of missing values per point) is also a low-degree polynomial as long as at least one of $k$ and $j$ is small. Actually, we justify in Theorem 1.2 that this exponential dependence in $\min\{j, k\}$ cannot be further improved, as long as the coreset size is in a similar parameter regime, i.e., the coreset size is of the form $f(j, k) \cdot \operatorname{poly}(\epsilon^{-1} d \log n)$. We provide the proof of Theorem 1.2 in the full version.

**Theorem 1.2.** *Consider the $k$-MEANS with missing values problem in $\mathbb{R}^d_?$ where each point can have at most $j$ missing coordinates. Assume there is an algorithm that constructs an $\epsilon$-coreset of size $f(j, k) \cdot \operatorname{poly}(\epsilon^{-1} d \log n)$, then $f(j, k)$ can not be as small as $2^{o(\min(j,k))}$.*

Furthermore, the space complexity of our construction algorithm is near-linear, and since our coreset is clearly mergeable, it is possible to apply the merge-and-reduce method [HM04] to convert our construction into a streaming algorithm of space $\operatorname{poly} \log n$. Prior to our result, the only known coreset construction for clustering with missing values is for the special case $j = 1$ [MF19][2] and has size $k^{O(k)} \cdot (\epsilon^{-2} d \log n)$. Since our coreset has size $\operatorname{poly}(k \epsilon^{-1} d \log n)$ when $j = 1$, it improves the dependence on $k$ over that of [MF19] by a factor of $k^{O(k)}$.

---

[2]In fact, [MF19] considers a slightly more general setting where the input are arbitrary lines that are not necessarily axis-parallel.

**Near-linear time PTAS for $k$-MEANS with missing values.** Very recently, a PTAS for $k$-MEANS with missing values, was obtained by Eiben, Fomin, Golovach, Lochet, Panolan, and Simonov [EFG$^+$21]. Its time bound is *quadratic*, namely $O(2^{\text{poly}(jk/\epsilon)} \cdot n^2 d)$, and since our coreset can be constructed in near-linear time, we can speedup this PTAS to *near-linear* time by first constructing our coreset and then running this PTAS on the coreset.

**Corollary 1.3** (Near-linear time PTAS for $k$-MEANS with missing values). *There is an algorithm that, given $0 < \epsilon < 1/2$, integers $d, j, k \geq 1$, and a set $X \subset \mathbb{R}^d$ of $n$ points each having at most $j$ missing values, it finds with constant probability a $(1 + \epsilon)$-approximation for $k$-MEANS on $X$, and runs in time $\tilde{O}\big((jk)^{O(\min\{j,k\})} \cdot nd + 2^{\text{poly}(jk/\epsilon)} \cdot d^{O(1)}\big)$.*

**Experiments.** We implement our algorithm and validate its performance on various real and synthetic data sets in Section 4. Our coreset exhibits flexible tradeoffs between coreset size and accuracy, and generally outperforms a uniform-sampling baseline and a baseline that is based on imputation, in both error rate and stability, especially when the coreset size is relatively small. In particular, on each data set, a coreset of moderate size 2000 (which is 0.5%-5% of the data sets) achieves low empirical error (5%-20%). We further demonstrate an application and use our coresets to accelerate a Lloyd's-style heuristic adapted to the missing-values setting. The experiments suggest that running the heuristic on top of our coresets gives equally good solutions (error $< 1\%$ relative to running on the original data set) but is much faster (speedup $> 5$x).

## 1.2 Technical Overview

Our coreset construction is based on the importance sampling framework introduced by Feldman and Langberg [FL11] and subsequently improved and generalized by [FSS20, BJKW21]. In the framework, one first computes an importance score $\sigma_x$ for every data point $x \in X$, and then draws independent samples with probabilities proportional to these scores. When no values are missing, the importance scores can be computed easily, even for general metric spaces [VX12b, FSS20, BJKW21]. However, a significant challenge with missing values is that distances do not satisfy the triangle inequality, hence importance scores cannot be easily computed.

We overcome this hurdle using a method introduced by Varadarajan and Xiao [VX12a] for projective clustering (where the triangle inequality similarly does not hold). They reduce the importance-score computation to the construction of a coreset for $k$-CENTER objective; this method is quite different from earlier approaches, e.g. [FL11, VX12b, FSS20, BJKW21], and yields a coreset for $k$-MEANS whose size depends linearly on $\log n$ and of course on the size of the $k$-CENTER coreset. (Mathematically, this arises from the sum of all importance scores.) We make use of this reduction, and thus focus on constructing (efficiently) a small coreset for $k$-CENTER with missing values.

An immediate difficulty is how to deal with the missing values. We show that it is possible to find a collection of subsets of coordinates $\mathcal{I}$ (so each $I \in \mathcal{I}$ is a subset of $[d]$), such that if we construct $k$-CENTER coresets $S_I$ on the data set "restricted" to each $I \in \mathcal{I}$, then the union of these $S_I$'s is a $k$-CENTER coreset for the original data set with missing values. Crucially, we ensure that each "restricted" data set does not contain any missing value, so that it is possible to use a classical coreset construction for $k$-CENTER. Finally, we show in a technical lemma how to find a collection as necessary of size $|\mathcal{I}| \leq (jk)^{O(\min\{j,k\})}$.

Since a "restricted" data set does not contain any missing values, we can use a classical $k$-CENTER coreset construction, and a standard construction has size $O(k\epsilon^{-d})$ [AP02], which is known to be tight. We bypass this $\epsilon^{-d}$ limitation by observing that actually $\tilde{O}(1)$-coreset for $k$-CENTER suffices, even though the final coreset error is $\epsilon$. We observe that an $\tilde{O}(1)$-coreset can be constructed using a variant of Gonzalez's algorithm [Gon85].

To implement Gonzalez's algorithm, a key step is to find the *furthest* neighbor of a given subset of at most $O(k)$ points, and a naive implementation of this runs in linear time, which overall yields a quadratic-time coreset construction, because the aforementioned reduction of [VX12a] actually requires $\Theta(n/k)$ successive runs of Gonzalez's algorithm. To resolve this issue, we propose a fully-dynamic implementation of Gonzalez's algorithm so that a furthest-point query is answered in time $\text{poly}(k \log n)$, and the point-set is updated between successive runs instead of constructed from scratch. Our dynamic algorithm is based on a random-projection method that was proposed for furthest-point queries in the streaming setting [Ind03]. Specifically, we project the (restricted) data

set onto several random directions, and on each projected (one-dimensional) data set we apply a data structure for intervals.

## 1.3 Additional Related Work

Coresets for $k$-MEANS and $k$-MEDIAN clustering have been studied extensively for two decades, and we only list a few notable results. The first strong coresets for Euclidean $k$-MEANS and $k$-MEDIAN were given in [HM04]. In the last decade, most work on coresets for clustering follows the importance sampling framework initiated in [LS10, FL11]. In Euclidean space, recent work showed that coresets for $k$-MEANS and $k$-MEDIAN clustering can have size that is independent of the Euclidean dimension [FSS20, SW18, HV20]. Beyond Euclidean space, coresets of size independent of the data-set size were constructed also for many important metric spaces [HJLW18, BJKW21, CASS21]. A more comprehensive overview can be found in recent surveys [Phi17, Fel20].

Recently, attention was given also to non-traditional settings of coresets for clustering, including coresets for Gaussian mixture models (GMM) [LFKF17, FKW19]; simultaneous coresets for a large family of cost functions that include both $k$-MEDIAN and $k$-CENTER [BJKW19]; and coresets for clustering under fairness constraints [HJV19]. Also considered were settings that capture uncertainty, for example when each point is only known to lie in a line (i.e., clustering lines) [MF19], and when each point comes from a finite set (i.e., clustering point sets) [JTMF20].

## 2 Preliminaries

We represent a data point as a vector in $(\mathbb{R} \cup \{?\})^d$, and a coordinate takes "?" if and only if it is missing. Let $\mathbb{R}_?^d$ be a shorthand for $(\mathbb{R} \cup \{?\})^d$. Throughout, we consider a data set $X \subset \mathbb{R}_?^d$. The distance is evaluated only on the coordinates that are present in both $x, y$, i.e.,

$$\forall x, y \in \mathbb{R}_?^d, \qquad \text{dist}(x, y) := \sqrt{\sum_{i: x_i, y_i \neq ?} (x_i - y_i)^2}.$$

For $x \in \mathbb{R}_?^d$, we denote the set of coordinates that are not missing by $I_x := \{i : x_i \neq ?\}$. For integer $m \geq 1$, let $[m] := \{1, \ldots, m\}$. For two points $p, q \in \mathbb{R}_?^d$ and an index set $I \subseteq I_p \cap I_q$, we define the *I-induced distance* to be $\text{dist}_I(p, q) := \sqrt{\sum_{i \in I} (p_i - q_i)^2}$. A point $x \in \mathbb{R}_?^d$ is called a $j$-point if it has at most $j$ missing coordinates, i.e., $|I_x| \geq d - j$.

We consider a general $k$-clustering problem called $(k, z)$-clustering, which asks to minimize the following objective function. This objective function (and problem) is also called $k$-MEDIAN when $z = 1$ and $k$-MEANS when $z = 2$.

**Definition 2.1** ($(k, z)$-CLUSTERING). For data set $X \subset \mathbb{R}_?^d$ and a center set $C \subset \mathbb{R}^d$ containing $k$ (usual) points, let

$$\text{cost}_z(X, C) := \sum_{x \in X} \text{dist}^z(x, C).$$

**Definition 2.2** ($\epsilon$-Coreset for $(k, z)$-CLUSTERING). For data set $X \subset \mathbb{R}_?^d$, we say a weighted set $S \subseteq X$ with weight function $w : S \to \mathbb{R}_+$ is an $\epsilon$-coreset for $(k, z)$-CLUSTERING, if

$$\forall C \subset \mathbb{R}^d, |C| = k, \qquad \sum_{x \in S} w(x) \cdot \text{dist}^z(x, C) \in (1 \pm \epsilon) \cdot \text{cost}_z(X, C).$$

## 3 Coresets

**Theorem 3.1.** *There is an algorithm that, given as input a data set $X \subset \mathbb{R}_?^d$ of size $n = |X|$ consisting of $j$-points and parameters $k, z \geq 1$ and $0 < \epsilon < 1/2$, constructs with constant probability an $\epsilon$-coreset of size $m = \tilde{O}\left(z^z \cdot \frac{(j+k)^{j+k+1}}{j^j k^{k-z-2}} \cdot \epsilon^{-2} (d \log n)^{\frac{z+2}{2}}\right)$ for $(k, z)$-CLUSTERING of $X$, and runs in time $\tilde{O}\left(\frac{(j+k)^{j+k+1}}{j^j k^{k-2}} \cdot nd + m\right)$.*

Theorem 3.1 is the main theorem of this paper, and we only present a sketch of the proof in this section due to the space limitation. Please see the full version for a more detailed and self-contained proof, as well as a complete description of our algorithm. We remark that $\frac{(j+k)^{j+k}}{j^j k^k} \leq (jk)^{O(\min(j,k))}$ which is used in the statement of Theorem 1.1.

As mentioned in Section 1, we use importance sampling method which is a well-known technique for constructing coresets [FL11, FSS20]. A key step is to compute for every data point $x \in X$ an importance score $\sigma_x \geq 0$ that estimates its maximum relative contribution to any solution. The computation of $\{\sigma_x\}$ is standard in metric spaces, see e.g. [FSS20, BJKW21], but this is not applicable for us because distances with missing values do not satisfy the triangle inequality. Hence, we employ an alternative approach proposed by Varadarajan and Xiao [VX12a, Lemma 3.1], which reduces the computation of $\{\sigma_x\}$ to finding coresets for $k$-CENTER. This coreset concept, adapted to our setting, is defined as follows.

**Definition 3.1.** An $\alpha$-*coreset for* $k$-CENTER of a data set $X \subset \mathbb{R}^d_?$ is a subset $Y \subseteq X$ such that

$$\forall C \subset \mathbb{R}^d, |C| = k, \qquad \max_{x \in X} \text{dist}(x, C) \leq \alpha \cdot \max_{y \in Y} \text{dist}(y, C).$$

We focus on an efficient construction of an $\tilde{O}(1)$-coreset for $k$-CENTER. The main concern is that the reduction in [VX12a, Lemma 3.1] requires constructing a $k$-CENTER coreset for multiple data sets. Fortunately, these data sets are related — each data set is a subset of the previous one — and thus to execute the reduction in near-linear time, we need a $k$-CENTER coreset construction that supports efficient point deletions. Such a dynamic coreset for $k$-CENTER with missing values is our main technical contribution. We stated it next, and outline its proof in Section 3.1.

**Lemma 3.2.** *There is a randomized dynamic algorithm with the following guarantees. The input is a dynamic set $X \subset \mathbb{R}^d_?$ of $j$-points, such that $X$ undergoes $q$ adaptive updates (point insertions and deletions) and the points ever added are fixed in advance (non-adaptively). The algorithm maintains in time $\tilde{O}\left(\frac{(j+k)^{j+k+1}}{j^j k^k} \cdot (j + k \log q)(d + k^2 \log q)\right)$ per update, a subset $Y \subseteq X$ of size $|Y| \leq O\left(\frac{(j+k)^{j+k+1}}{j^j k^{k-1}} \cdot \log d\right)$ such that with constant probability, $Y$ is an $O(k\sqrt{d \log q})$-coreset for $k$-CENTER on $X$ after every update.*

### 3.1  Proof of Lemma 3.2: Dynamic $\tilde{O}(1)$-Coresets for $k$-Center Clustering

As mentioned, the high level idea is to identify a collection $\mathcal{I}$ of subsets of coordinates (so each $I \in \mathcal{I}$ satisfies $I \subseteq [d]$), construct for each $I_i \in \mathcal{I}$ an $\alpha$-coreset $Y_i$ (for $\alpha$ determined later) for $k$-CENTER on the data set $X$ with coordinates *restricted* on $I_i$ (as defined below), and then their union $\bigcup_i Y_i$ would be the overall $\alpha\sqrt{d}$-coreset for $k$-CENTER on $X$.

**Definition 3.2.** For a point $p \in \mathbb{R}^d_?$ and a subset $I \subseteq I_p$, define $p_{|I} \in \mathbb{R}^I$ in the obvious way, by selecting the coordinates $\{p_i\}_{i \in I}$. Define the *$I$-restricted data set* to be $X_{|I} := \{p_{|I} : p \in X, I \subseteq I_p\}$. Since each vector in $X_{|I}$ arises from a specific vector in $X$, a subset $Y \subseteq X_{|I}$ corresponds to a specific subset of $X$, and we shall denote this subset by $Y^{-1}$.

We observe that $X_{|I} \subset \mathbb{R}^{|I|}$, namely, has no missing values (because of the condition $I \subseteq I_p$). Thus, the metric space on the restricted data set is an ordinary metric space that satisfies the triangle inequality, and so our goal is reduced to constructing $k$-CENTER coresets for this ordinary setting. However, another key step is to identify a small collection $\mathcal{I}$ such that the union of the coresets restricted on $\mathcal{I}$ yields a coreset. To this end, we consider the so-called $(j, k, d)$-family of coordinates as in Definition 3.3. We show in Lemma 3.3 that such a family guarantees the correctness of the coreset, and in Lemma 3.4 that a small family exists and moreover can be constructed efficiently.

**Definition 3.3.** A collection of sets $\mathcal{I} \subset 2^{[d]}$ is called a $(j, k, d)$-*family* if for every two disjoint subsets $J, K \subset [d], |J| = j, |K| = k$, the family includes $I \in \mathcal{I}$ that misses $J$ and contains $K$, i.e., $I \cap J = \emptyset$ and $K \subset I$.

**Lemma 3.3.** *Suppose $\mathcal{I}$ is a $(j, k, d)$-family Let $X \subseteq \mathbb{R}^d_?$ be a set of $j$-points, and for every $I \in \mathcal{I}$, let $Y_I$ be an $\alpha$-coreset for $k$-CENTER on $X_{|I}$. Then $\cup_{I \in \mathcal{I}} Y_I^{-1}$ is an $\alpha\sqrt{d}$-coreset for $k$-Center on $X$.*

*Proof.* It suffices to show that for any center set $C = \{c^1, \ldots, c^k\} \subseteq \mathbb{R}^d$ with $k$ points and $x \in X$, if $\text{dist}(x, C) \geq r$ for some $r \geq 0$, then we can find a coreset point $y \in \cup_{I \in \mathcal{I}} Y_I^{-1}$ such that $\text{dist}(y, C) \geq \frac{r}{\alpha\sqrt{d}}$.

For $i \in [k]$, let $t_i \in \arg\max_{t \in I_x} |x_t - c_t^i|$, i.e., $t_i$ is the index of coordinate that contributes the most in distance $\text{dist}(x, c^i)$, so $|x_{t_i} - c_{t_i}^i| \geq \frac{r}{\sqrt{d}}$. Let $K$ be any $k$-subset such that $K \subseteq I_x$ and $\{t_1, \ldots, t_k\} \subseteq K$. Since $\mathcal{I}$ is a $(j, k, d)$-family and $|I_x| \geq d - j$, by definition, there exists an $I \subseteq \mathcal{I}$ such that $K \subseteq I \subseteq I_x$. We note that

$$\text{dist}(x_{|I}, C_{|I}) = \text{dist}_I(x, C) = \min_{i \in [k]} \text{dist}_I(x, c^i) \geq \min_{i \in [k]} \text{dist}_K(x, c^i) \geq \min_{i \in [k]} |x_{t_i} - c_{t_i}^i| \geq \frac{r}{\sqrt{d}}.$$

Since $I \subseteq I_x$, we know that $x_{|I} \in X_{|I}$. As $Y_I$ is an $\alpha$-coreset for $X_{|I}$, we know that there exists $y \in Y_I^{-1}$ such that

$$\text{dist}(y, C) \geq \text{dist}_I(y, C) = \text{dist}(y_{|I}, C_{|I}) \geq \frac{\text{dist}(x_{|I}, C_{|I})}{\alpha} \geq \frac{r}{\alpha\sqrt{d}}.$$

$\square$

Lemma 3.4 asserts the existence of a small $(j, k, d)$-family. We remark that this combinatorial structure has been employed in designing fault-tolerant data structures and algorithms (cf. [DK11, DGR21, KP21]), and they obtained similar bounds although their context and language is different.

**Lemma 3.4.** *There is a $(j, k, d)$-family $\mathcal{I}$ of size $|\mathcal{I}| = O\left(\frac{(j+k)^{j+k+1}}{j^j k^k} \log d\right)$. Moreover, there is a randomized algorithm that constructs such $\mathcal{I}$ in time $O(d \cdot |\mathcal{I}|)$ with probability at least $1 - \frac{1}{d^{j+k}}$.*

$k$-**CENTER coreset for restricted data set via Gonzalez's algorithm.** Finally, the $k$-CENTER coreset for the restricted data set on each $I \in \mathcal{I}$ is constructed using an approximate version of Gonzalez's algorithm [Gon85], where we first pick an arbitrary data point as the initial coreset, and in every iteration an $\tilde{O}(1)$-approximate furthest neighbor in the dataset is picked into the coreset, and we do this for $k$ times. We show that the resulting coreset, consisting of $k + 1$ points, is an $\tilde{O}(1)$-coreset for $k$-CENTER. The assumption that the input forms a metric space is crucial, and this is guaranteed since we always run on a restricted data set that satisfies the triangle inequality.

**Dynamic implementation of Gonzalez's algorithm.** To make this $k$-CENTER coreset construction dynamic, we adapt the random projection technique to Gonzalez's algorithm. We project the (restricted) point set onto several random lines and use a one-dimensional data structure to construct $k$-CENTER coreset for each of these (projected) one-dimensional data set. Note that the key step in Gonzalez's algorithm is finding a furthest neighbor, and we can show that our projection method yields an $O(k\sqrt{\log n})$-approximation of the furthest neighbor with high probability.

**Lemma 3.5.** *Let $A \subset \mathbb{R}^d$, $|A| = n$, $\delta > 0$ and integer $k \geq 1$. Let $\mathcal{V}$ be a collection of $O(k \log n + \log \frac{1}{\delta})$ random vectors, each drawn independently from $\mathcal{N}(0, I_d)$. Then with probability at least $1 - \delta$, for every $P \subseteq A$ and $Q \subseteq A$, $|Q| \leq k$, if $p^\star$ is a furthest point in $P$ from $Q$, and for every $v \in \mathcal{V}$ we let $\langle p^v, v \rangle$ be a furthest point in $\langle P, v \rangle$ from $\langle Q, v \rangle$, then*

$$\text{dist}(p^\star, Q) \leq O(k\sqrt{\log n}) \cdot \max_{v \in \mathcal{V}} \text{dist}(p^v, Q),$$

*where we denote $\langle X, v \rangle := \{\langle x, v \rangle : x \in X\}$.*

For each one-dimensional line, we use a balanced search tree structure to support the point update and the furthest neighbor query. All operations can be done in $O(k \log m)$ time where $m$ is the number of currently inserted elements. This combining with the above lemmas implies Lemma 3.2.

# 4 Experiments

We implement our proposed coreset construction algorithm, and we evaluate its performance on real and synthetic datasets. We focus on $k$-MEANS with missing values, and we examine the speedup

Table 1: Parameters of the datasets. $n$ is the number of data points, $d$ is the dimension, $k$ is the number of clusters, $j$ is the maximum number of missing coordinates for each point. $n,d,j$ are given, and $k$ is chosen by us.

| Data set | $n$ | $d$ | $k$ | $j$ |
|---|---|---|---|---|
| Russian housing | 30471 | 4 | 3 | 3 |
| KDD cup | 50000 | 31 | 5 | 30 |
| Vertical farming | 400180 | 4 | 2 | 4 |
| Synthetic | 200000 | 3 | 3 | 3 |

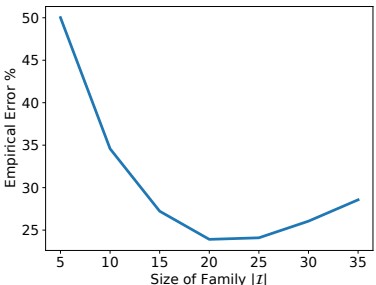

Figure 1: The average empirical error of Russian housing data set with respect to family size $|\mathcal{I}|$ on 10 independent experiments.

for a Lloyd's-style heuristic. In addition to measuring the absolute performance of our coreset, we also compare it with a) uniform sampling baseline, which is a naive way to construct coresets, and b) an imputation-based baseline where missing values are filled in by random values and then a standard importance-sampling coreset construction (cf. [FL11]) is run on top of it. We implement the algorithms using C++ 11, on a laptop with Intel i5-8350U CPU and 8GB RAM.

**Datasets.** We run our experiments on three real datasets and one synthetic dataset. Below, we briefly describe how we process and choose the attributes of the dataset, and the parameters of the datasets after processing are summarized in Table 1.

1. Russian housing [Rus17] is a dataset on Russian house market. We pick four main numerical attributes of the houses which are the full area, the live area, the kitchen area and the price, and the price attribute is divided by $10^5$ so as it lies in the similar range of other attributes. Three columns regarding area contain missing values, and the price column doesn't contain any missing value.
2. KDDCup 2009 [KDD09] is a dataset on customer relationship prediction. We pick 31 numerical attributes that have similar magnitudes. Each column contains missing values.
3. Vertical farming [Sam21] is a dataset about cubes which are used for advanced vertical farming. We include all of four numerical attributes of the dataset. Each column contains missing values.
4. Synthetic dataset. We generate a large synthetic dataset to validate our algorithm's scalability. Data points are randomly generated so that $97\%$ of them are in a square and $3\%$ of them are far away from the square. After that, we delete $25\%$ of attributes at random. We remark that the $3\%$ far away points is to make the dataset less uniform which prevents it from being trivial for clustering.

**Implementation notes.** In our experiments, we follow a standard practice of fixing coreset size in each experiment (cf. [BBH$^+$20, JTMF20]). Recall that when computing the importance score, our algorithm chooses a family $\mathcal{I}$ of subsets of coordinates and work on each restricted data set $X_{|I}$ for $I \in \mathcal{I}$ (see Section 3.1). For a fixed size coreset, the family size $|\mathcal{I}|$ is a parameter that needs to be optimized. In Figure 1, we plot the empirical error (defined in (1), Section 4.1) for the Russian housing dataset with respect to the family size $|\mathcal{I}|$. Although Lemma 3.4 gives a theoretical upper bound on $|\mathcal{I}|$ but our experiments suggest that a much smaller size $|\mathcal{I}| = 20$ is optimal in this case.

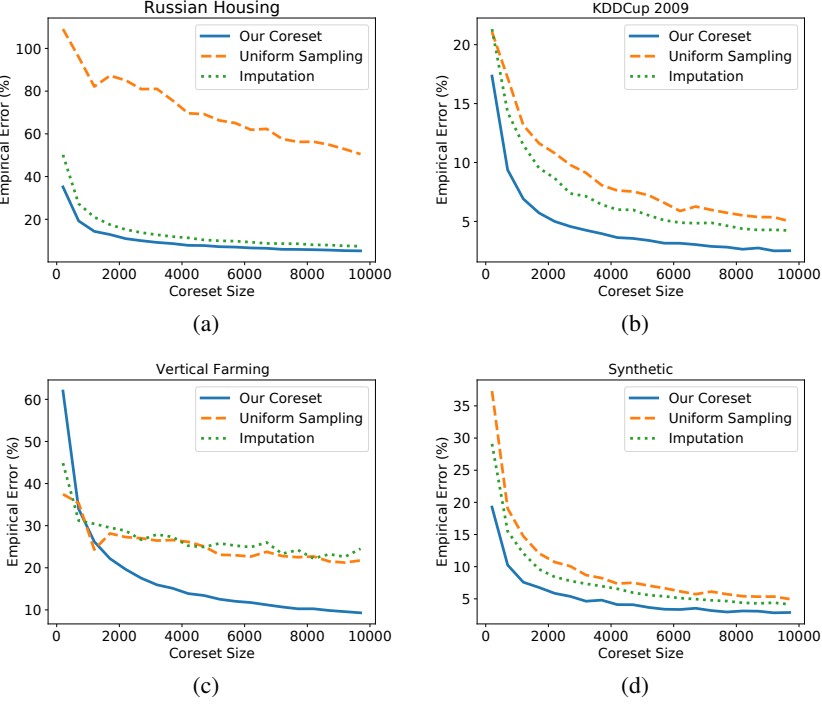

Figure 2: Accuracy evaluation for the datasets with respect to varing coreset sizes, compared against uniform sampling and imputation baselines.

## 4.1 Accuracy of Coresets

We evaluate the accuracy versus size tradeoff of our coresets. Since the coreset should preserve the clustering cost for *all* centers, we evaluate the accuracy by testing the *empirical error* on a selected set of centers $\mathcal{C}$. Namely, for a data set $X$, a coreset $D \subseteq X$ and a collection of center sets $\mathcal{C}$, we define the empirical error of $D$ as

$$\text{err}(D) = \max_{C \in \mathcal{C}} \frac{|\text{cost}(D, C) - \text{cost}(X, C)|}{\text{cost}(X, C)}. \tag{1}$$

We use a randomly selected collection of centers $\mathcal{C}$ that consists of 100 randomly generated $k$-subset $C \subset \mathbb{R}^d$. Since both the evaluation method and the algorithm has randomness, we run the experiment for $T = 10^3$ times with independent random bits and report the average empirical error to make it stable. We choose 20 different coreset sizes from 200 to 9700 in a step size of 500, and report the corresponding average empirical error.

**Results.** We report the size versus accuracy tradeoff of our coreset for all four datasets in Figure 2, and record the standard deviation in Figure 3. We compare these results against the abovementioned uniform sampling and imputation baseline. As can be seen from the figures, the accuracy of our coreset improves when the size increases, and we achieve $5\%$-$20\%$ error using only 2000 coreset points (which is within $0.5\% - 5\%$ of the datasets). This $5\%$-$20\%$ error is likely to be enough for practical use, since practical algorithms for $k$-MEANS are approximation algorithms anyway. Our coresets generally outperform both the uniform sampling and imputation baselines on almost every coreset sample size, and the advantage is more significant when the coreset size is relatively small. Moreover, our coresets have a much lower variance.

## 4.2 Speedup of Lloyd's-style Heuristic

Coresets often help to speed up existing approximation algorithms. Before our work, the only algorithm for $k$-MEANS with provable guarantees for multiple missing values was [EFG+21]. Unfortunately, [EFG+21] is not practical even when combined with coresets, since it contains several

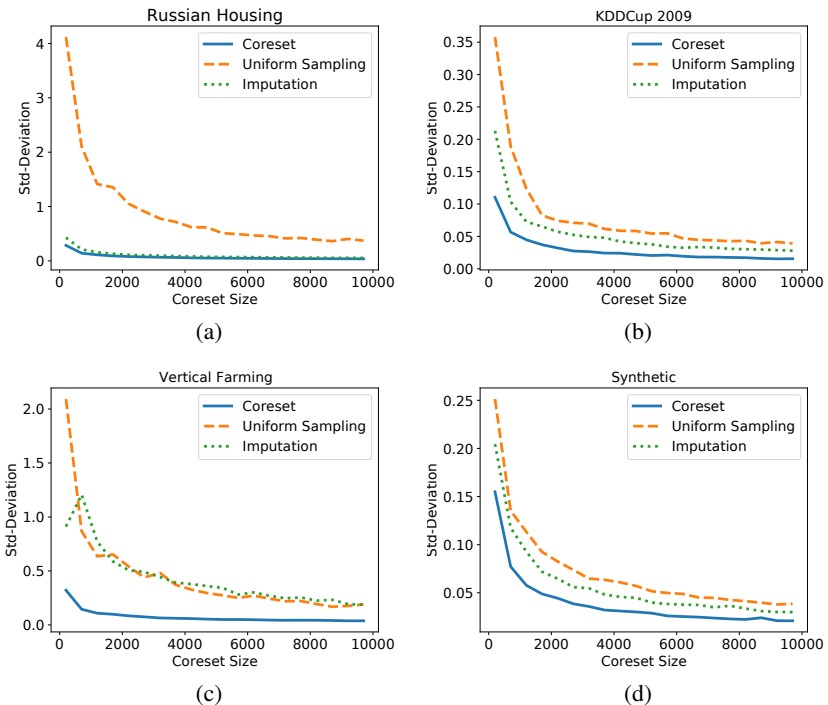

Figure 3: Standard deviation for the size-error evaluation.

enumeration procedures that require $\Theta(\exp(\text{poly}(\epsilon^{-1}jk)))$ time. We consider a variant of Lloyd's heuristic [Llo82] that is adapted to the missing-value setting, and we evaluate its speedup with coresets. The algorithm is essentially the same as the original Lloyd's algorithm, except that the distance as well as the optimal 1-mean for a cluster (which can be computed optimally in $O(d|P|)$ for a cluster $P$ [EFG+21]), is computed differently. We show that our coreset can significantly accelerate this algorithm. In particular, we run the modified Lloyd's heuristic directly on the original dataset, and take its running time and objective value as the comparison reference. Then we run this modified Lloyd's heuristic again, but on top of our coreset and the uniform sampling baseline respectively, and we compare both the speedup and the relative error[3] against the reference. The experiments are run on the Russian housing data set where the number of iterations of the modified Lloyd's is set to $T = 5000$ and the number of clusters is set to a small value $k = 3$ so as the heuristic is likely to find a local minimum faster. Again, to obtain a stable result, we run the experiments for $40$ times with independent random bits and report the average relative errors and running time.

**Results.** The relative error with respect to varying coreset sizes can be found in Figure 4a. We can see that the relative error of Lloyd's algorithm running on our coreset is consistently low, while the uniform sampling baseline has several times higher error and the error does not seem to improve even when improving the size. We note that relative errors for both our coreset and uniform sampling are significantly lower than that we observe from the empirical error in Figure 2a. In fact, they are not necessarily comparable since the empirical error in Figure 2a is always evaluated on a same center, while what we compare in Figure 4a is the center sets found by the modified Lloyd's running on different data sets. This also helps to explain why improving the size of uniform sampling may not result in a better solution, since as shown in Figure 2a, uniform sampling has a large empirical error (around $50\%$), so a good solution for the uniform sample may not be a good solution for the original data set.

The running time of the modified Lloyd's on top of our coresets can be found in Figure 4b, and the running time of Lloyd's on the original dataset is 22.9s (which is not drawn on the figure). To make a fair comparison, we also take the coreset construction time into account. Note that coreset size is not a dominating factor in the running time of coreset construction, since the majority of time is

---

[3]For $x \in \mathbb{R}_+$, the relative error of $x$ against a reference $x^\star > 0$ is defined as $\frac{|x-x^\star|}{x^\star}$.

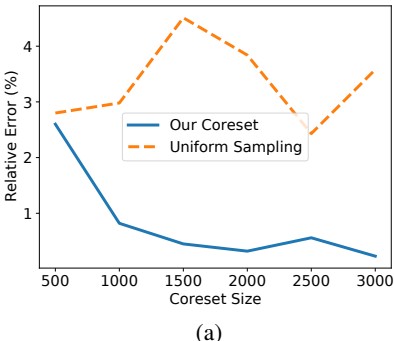
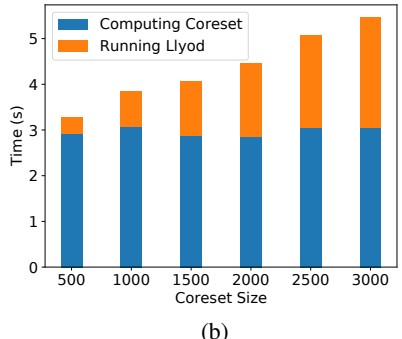

$$(a) \qquad\qquad\qquad\qquad (b)$$

Figure 4: Relative error and running time evaluation for the Lloyd's heuristic on the coreset, with respect to varying coreset sizes. The left figure demonstrates the relative error, and the right figure shows the running time of constructing our coreset, and the time for the modified Lloyd's heuristic running on top of our coreset.

spent on computing the importance scores and the coreset size only affects the number of samples. A coreset of size only $1000$ can achieve $< 1\%$ error, and the running time of constructing the coreset and applying Lloyd's on top of it are $3$s and $0.8$s, respectively, which offers more than $5$ times of speedup. We remark that our experiments only demonstrate the speedup in a single-machine scenario, and the speedup will increase in the parallel or distributed setting.

## 5 Conclusion

Our coreset construction builds upon the sensitivity-sampling method (cf. [FL11]). However, a central technical challenge is that the standard method to compute the sensitivity scores breaks, because distances between points with missing values do not satisfy the triangle inequality. We overcome this using another known method, of [VX12a], that requires a coreset for $k$-CENTER. Our main innovation is a near-linear time algorithm that computes an $O(1)$-approximate $k$-CENTER coreset for points with missing values. To this end, we need the following key steps, which constitute our main technical contribution.

- We reduce the $k$-CENTER coreset construction with missing values, to the construction of traditional $k$-CENTER coresets (i.e., without missing values) on a series of instances. These instances are built by restricting data points with missing values to a carefully-chosen collection of subspaces. The guarantee needed from this collection is a certain combinatorial structure, and we indeed prove it exists.

- The method of Varadarajan and Xiao executes the $k$-CENTER coreset algorithm many times, and overall takes quadratic time. To improve the running time, we design an efficient dynamic algorithm for the well-known Gonzales' algorithm (which computes an $O(1)$-approximate $k$-CENTER coreset). The main idea in this dynamic algorithm is to project the data points onto (data-oblivious) random 1D lines, and build on each line a dynamic data structure that supports furthest-neighbor queries (in 1D).

Finally, we implemented our algorithm and the experiments indicate that our algorithm is efficient and accurate enough to be potentially applicable in practice.

**Future directions.** As an immediate follow-up, one could try to improve our coreset size, e.g., removing the dependence in $\log n$. Our input can be viewed as axis-parallel affine-subspaces. Hence, another an interesting direction is to obtain coresets for the more general setting where the input consists of general affine-subspaces.

**Potential negative societal impacts.** Our paper focuses on computational issues (improving time and space) of known clustering tasks. Clustering methods in general have potential issues with fairness and privacy, which applies also to our work, but our research is not expected to introduce new negative societal impact beyond what is already known.

## Acknowledgments and Disclosure of Funding

The majority of this work was done when Shaofeng Jiang was at Aalto University. This work is partially supported by ONR Award N00014-18-1-2364, by the Israel Science Foundation grant #1086/18, by a Weizmann-UK Making Connections Grant, and by a Minerva Foundation grant.

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
