# Appendices

## A  Full Version of Section 3

This section is the full version of Section 3. We restate Theorem 3.1 in Theorem A.1 which is the main theorem. Lemma 3.2, Lemma 3.3, and Lemma 3.4 correspond to Lemma A.6, Lemma A.7, and Lemma A.8, respectively. Lemma 3.5 is not used explicitly in the full version but we present it in Section 3 for the sake of presentation, sicne it captures the main idea of A.12 as well as the a central part of the proof for Lemma A.13.

In addition, we present all the missing details of the importance sampling framework (Lemma A.2), the reduction of Varadarajan and Xiao [VX12a] (Lemma A.5), and the Gonzales's algorithm as well as its dynamic implementation (Lemma A.13).

**Theorem A.1.** *There is an algorithm that, given as input a data set $X \subset \mathbb{R}^d_?$ of size $n = |X|$ consisting of $j$-points and parameters $k, z \geq 1$ and $0 < \epsilon < 1/2$, constructs with constant probability an $\epsilon$-coreset of size $m = \tilde{O}\left(z^z \cdot \frac{(j+k)^{j+k+1}}{j^j k^{k-z-2}} \cdot \frac{(d \log n)^{\frac{z+2}{2}}}{\epsilon^2}\right)$ for $(k, z)$-CLUSTERING of $X$, and runs in time $\tilde{O}\left(\frac{(j+k)^{j+k+1}}{j^j k^{k-2}} \cdot nd + m\right)$.*

We remark that $\frac{(j+k)^{j+k}}{j^j k^k} = (jk)^{O(\min(j,k))}$. To see this, assume $j \geq k$ w.l.o.g., so $\frac{(j+k)^j}{j^j} = (1 + \frac{k}{j})^j \leq e^{\frac{k}{j} \cdot j} = e^k$ and $\frac{(j+k)^k}{k^k} \leq (j+k)^k$.

Theorem A.1 is the main theorem of this paper, and we present the proof in this section. As mentioned in Section 1, the coreset is constructed via importance sampling, by following three major steps.

1. For each data point $x \in X$, compute an importance score $\sigma_x \geq 0$.
2. Draw $N$ (to be determined later) independent samples from $X$, such that $x \in X$ is sampled with probability $p_x \propto \sigma_x$.
3. Denote the sampled (multi)set as $S$, and for each $x \in S$ define its weight $w(x) := \frac{1}{p_x N}$. Report the weighted set $S$ as the coreset.

The importance score $\sigma_x$ is usually defined as (an approximation) of the *sensitivity* of $x$, denoted

$$\sigma_x^\star := \sup_{C \subset \mathbb{R}^d, |C| = k} \frac{\text{dist}^z(x, C)}{\text{cost}_z(X, C)}, \tag{2}$$

which measures the maximum possible relative contribution of $x$ to the objective function.

Usually, there are two main challenges with this approach. First, the sensitivity (2) is not efficiently computable because it requires to optimize over all $k$-subsets $C \subset \mathbb{R}^d$. Second, one has to determine the number of samples $N$ (essentially the coreset size) based on a probabilistic analysis of the event that $S$ is a coreset. Prior work on coresets has studied these issues extensively and developed a general framework, and we shall use the variant stated in Theorem A.2 below. This framework only needs an approximation to the sensitivities $\{\sigma_x^\star\}_{x \in X}$, more precisely it requires overestimates $\sigma_x \geq \sigma_x^\star$ whose sum $\sum_{x \in X} \sigma_x$ is bounded. Moreover, it relates the number of samples $N$ to a quantity called the *weighted shattering dimension* $\text{sdim}_{\max}$, which roughly speaking measures the complexity of a space (set of points) by the number of distinct ways that metric balls can intersect it. The definition below has an extra complication of a point weight $v$, which originates from the weight in the importance sampling procedure, and thus we need a uniform upper bound, denoted $\text{sdim}_{\max}$, over all possible weights.[4]

**Definition A.1** (Shattering dimension). Given a *weight* function $v : \mathbb{R}^d_? \to \mathbb{R}_+$, let $\text{sdim}_v(\mathbb{R}^d_?)$ be the smallest integer $t$ such that

$$\forall H \subset \mathbb{R}^d_?, |H| \geq 2 \qquad \left|\left\{B_v^H(c, r) : c \in \mathbb{R}^d, r \geq 0\right\}\right| \leq |H|^t,$$

where $B_v^H(c, r) := \{x \in H : v(x) \cdot \text{dist}(x, c) \leq r\}$. Let $\text{sdim}_{\max}(\mathbb{R}^d_?) := \sup_{v:\mathbb{R}^d_? \to \mathbb{R}_+} \text{sdim}_v(\mathbb{R}^d_?)$.

---

[4]In principle, this uniform upper bound is not necessary, and an upper bound for weights corresponding to the importance score suffices, but a uniform upper bound turns out to be technically easier to deal with.

Strictly speaking, Theorem A.2 has been proposed and proved only for metric spaces, but the proof is applicable also in our setting (where $\mathrm{dist}$ need not satisfy the triangle inequality), because it only concerns the *binary* relation between data points and center points (without an indirect use of a third point, e.g., by triangle inequality.)

**Theorem A.2** ([FSS20][5]). *Let $X \subset \mathbb{R}^d_?$ be a data set, and let $k, z \geq 1$. Consider the importance sampling procedure with importance scores that satisfy $\sigma_x \geq \sigma_x^\star$ for all $x \in X$, and with a sufficiently large number of samples*

$$N = \tilde{O}\left( \epsilon^{-2} k z^z \, \mathrm{sdim}_{\max}(\mathbb{R}^d_?) \sum_{x \in X} \sigma_x \right).$$

*Then with constant probability it reports an $\epsilon$-coreset for $(k, z)$-CLUSTERING.*

*Proof of Theorem A.1.* Because of Theorem A.2, it suffices to bound $\mathrm{sdim}_{\max}(\mathbb{R}^d_?)$, and to provide an efficient algorithm to estimate $\sigma_x$ whose sum is bounded. These two components are provided in Lemma A.3 and Lemma A.4 stated below (their proofs appear in Sections A.1 and A.2), Plugging these two lemmas into Theorem A.2, the main theorem follows. We provide an outline for the complete algorithm in Algorithm 1. $\qquad\square$

---

**Algorithm 1** Main algorithm

---

1: run Algorithm 3 to obtain $\sigma_x$ for $x \in X$
2: draw $N := \tilde{O}\left( z^z \cdot \frac{(j+k)^{j+k+1}}{j^j k^{k-z-2}} \cdot \frac{(d \log n)^{\frac{z+2}{2}}}{\epsilon^2} \right)$ independent samples $S$ from $X$, where $x \in X$ is
    sampled with probability $p_x \propto \sigma_x$
3: for $x \in S$, define weight $w(x) \leftarrow \frac{1}{p_x N}$
4: return weighted set $S$ with weight $w$ as the coreset

---

**Lemma A.3** (Shattering dimension bound). $\mathrm{sdim}_{\max}(\mathbb{R}^d_?) = O(d)$.

**Lemma A.4.** *There is an algorithm that, given a data set $X \subset \mathbb{R}^d_?$ of $n$ $j$-points, for $(k, z)$-CLUSTERING computes importance scores $\{\sigma_x\}_{x \in X}$ such that with constant probability,*

- $\sigma_x \geq \sigma_x^\star$ *for all $x \in X$; and*

- $\sum_{x \in X} \sigma_x \leq O\left( \frac{(j+k)^{j+k+1}}{j^j k^{k-z-1}} \cdot \sqrt{d^z \cdot \log^{z+2} n} \right)$,

*and its running time is $\tilde{O}\left( \frac{(j+k)^{j+k+2}}{j^j k^{k-2}} \cdot nd \right)$.*

## A.1 Proof of Lemma A.3: Shattering Dimension of $\mathbb{R}^d_?$

We now prove Lemma A.3, which asserts that $\mathrm{sdim}_{\max}(\mathbb{R}^d_?) = O(d)$. We remark that the shattering dimension bound for $\mathbb{R}^d$ without missing values has been proved in [FL11, Lemma 16.1] and our proof is actually an extension of it.

*Proof of Lemma A.3.* Let us verify Definition A.1. Consider $H \subset \mathbb{R}^d_?$ and a weight function $v : \mathbb{R}^d_? \to \mathbb{R}_+$. Recall that given $c \in \mathbb{R}^d$ and $r \geq 0$, we have $B_v^H(c, r) = \{h \in H : v(h) \cdot \mathrm{dist}(h, c) \leq r\}$ and $\mathrm{dist}(h, c)^2 = \sum_{i \in I_h} (h_i - c_i)^2$ for $h \in H$. We need to show that

$$\left| \{ B_v^H(c, r) : c \in \mathbb{R}^d, r \geq 0 \} \right| \leq |H|^{O(d)}. \tag{3}$$

Observe that

$$h \in B_v^H(c, r) \iff v(h) \cdot \mathrm{dist}(h, c) \leq r \iff -r^2 + \sum_{i \in I_h} (v^2(h)h_i^2 + v^2(h)c_i^2 - 2v^2(h)h_i c_i) \leq 0.$$

---

[5]Our theorem statement is based on [FSS20, Theorem 31], adapted to our context. One difference is that their theorem is about VC-dimension, but it is also applicable for shattering dimension. Another difference is that we use a more direct terminology that is specialized to metric balls in $\mathbb{R}^d_?$ instead of a general range space.

Next, we write this inequality in an alternative way, that separates terms depending $h$ from those depending on $c$ and $r$, more precisely as an inner-product $\langle f(h), g(c,r) \rangle \leq 0$ for vectors $f(h), g(c,r) \in \mathbb{R}^{3d+1}$. Now consider $f : H \to \mathbb{R}^d \times \mathbb{R}^d \times \mathbb{R}^d \times \mathbb{R}$ and $g : \mathbb{R}^d \times \mathbb{R} \to \mathbb{R}^d \times \mathbb{R}^d \times \mathbb{R}^d \times \mathbb{R}$ such that $f(h) = (p, q, t, -1)$, where $p, q, t \in \mathbb{R}^d$ and for $i \in [d]$

$$p_i = \begin{cases} v^2(h) \cdot h_i^2 & \text{if } i \in I_h \\ 0 & \text{otherwise} \end{cases} \quad q_i = \begin{cases} v^2(h) & \text{if } i \in I_h \\ 0 & \text{otherwise} \end{cases} \quad t_i = \begin{cases} -2v^2(h) \cdot h_i & \text{if } i \in I_h \\ 0 & \text{otherwise} \end{cases}$$

and $g(c,r) = (y, z, w, r^2)$, where $y, z, w \in \mathbb{R}^d$, $y_i = 1$, $z_i = c_i^2$, $w_i = c_i$ for $i \in [d]$. Then we have

$$h \in B_v^H(c,r) \iff \langle f(h), g(c,r) \rangle \leq 0.$$

For a vector $t \in \mathbb{R}^{3d+1}$, let $\text{proj}_-^H(t) := \{h \in H : \langle f(h), t \rangle \leq 0\}$ be the subset of $H$ that has nonpositive inner-product with $t$ (it can be viewed also as projection or a halfspace). Therefore, by (3), we have

$$\left| \{B_v^H(c,r) : c \in \mathbb{R}^d, r \geq 0\} \right| = \left| \{\text{proj}_-^H(g(c,r)) : c \in \mathbb{R}^d, r \geq 0\} \right| \leq \left| \{\text{proj}_-^H(t) : t \in \mathbb{R}^{3d+1}\} \right|.$$

We observe that

$$\left| \{\text{proj}_-^H(t) : t \in \mathbb{R}^{3d+1}\} \right| \leq |H|^{O(d)},$$

since this may be related to the shattering dimension of halfspaces in $\mathbb{R}^{3d+1}$, which is $O(d)$ and is a well-known fact in the PAC learning theory (cf. [vH14, Chapter 7.2]). This concludes the proof of Lemma A.3. $\qquad \square$

## A.2   Proof of Lemma A.4: Estimating Sensitivity Efficiently

We use a technique introduced by Varadarajan and Xiao [VX12a] that reduces the sensitivity-estimation problem to the problem of constructing a coreset for $k$-CENTER clustering. This coreset concept is defined as follows.

**Definition A.2.** An $\alpha$-*coreset for* $k$-CENTER of a data set $X \subset \mathbb{R}_?^d$ is a subset $Y \subseteq X$ such that

$$\forall C \subset \mathbb{R}^d, |C| = k, \qquad \max_{x \in X} \text{dist}(x, C) \leq \alpha \cdot \max_{y \in Y} \text{dist}(y, C).$$

Note that the error parameter $\alpha$ represents a multiplicative factor, which is slightly different from that of $\epsilon$ in $\epsilon$-coreset for $(k,z)$-CLUSTERING, and roughly corresponds to $\alpha = 1 + \epsilon$. The reasoning is that $\max_{y \in Y} \text{dist}(y, C)$ for $Y \subseteq X$ is always no more than $\max_{x \in X} \text{dist}(x, C)$, and therefore we only need to measure the contraction-side error.

The reduction in Lemma A.5 was presented in [VX12a], and we restate its algorithmic steps in Algorithm 2. This needs access to some Algorithm $\mathcal{A}$ that constructs an $\alpha$-coreset for $k$-CENTER on a point set $X \subset \mathbb{R}_?^d$. Each iteration $i$ calls Algorithm $\mathcal{A}$ to construct a $k$-CENTER coreset for the current point set $X$ (which is initially the entire data set), assign sensitivity estimates $O(\alpha^z / i)$ to every coreset point, and then remove these coreset points from $X$. These iterations are repeated until $X$ is empty.

---

**Algorithm 2** Sensitivity estimation from [VX12a, Lemma 3.1] for data set $X \subset \mathbb{R}_?^d$

**Require:** algorithm $\mathcal{A}$ that constructs $\alpha$-coreset for $k$-CENTER
 1: $i \leftarrow 1$
 2: **while** $X \neq \emptyset$ **do**
 3:     $P \leftarrow \mathcal{A}(X)$
 4:     **for** $x \in P$ **do**
 5:         $\sigma_x \leftarrow O(\alpha^z / i)$
 6:     **end for**
 7:     $X \leftarrow X \setminus P$
 8:     $i \leftarrow i + 1$
 9: **end while**

---

**Lemma A.5** ([VX12a, Lemma 3.1]). *Suppose algorithm $\mathcal{A}$ constructs an $\alpha$-coreset of size $T = T(\alpha, d, j, k)$ for $k$-CENTER an input $X \subset \mathbb{R}_?^d$. Then Algorithm 2 (which makes calls to this Algorithm $\mathcal{A}$) computes sensitivities $\{\sigma_x\}$ for $(k, z)$-CLUSTERING satisfying that $\sigma_x \geq \sigma_x^\star$ for all $x \in X$, and $\sum_{x \in X} \sigma_x \leq \alpha^z \cdot T \log |X|$.*

However, there are two outstanding technical challenges. First, there is no known construction of a small $k$-CENTER coreset for our clustering with missing values setting. Moreover, as can be seen from Algorithm 2, this reduction executes the $k$-CENTER coreset construction $\frac{|X|}{T}$ times (where $T$ is the size of the coreset as in Lemma A.5), and when using a naive implementation of the $k$-CENTER coreset construction, which naturally requires $\Omega(|X|)$ time, results overall in quadratic time, which is not very efficient.

First, to deal with question marks, we employ a certain family $\mathcal{I}$ of subset of coordinates (so each $I \in \mathcal{I}$ is a subset of $[d]$), and we *restrict* the data set $X$ on each $I \in \mathcal{I}$. Each restricted data set (restricted on some $I$) may be viewed as a data set in $\mathbb{R}^I$, without any question marks. We show that the union of $k$-CENTER coresets on all restricted data sets with respect all to $I \in \mathcal{I}$, forms a valid $k$-CENTER coreset for $X$ (which has question marks), provided that the family $\mathcal{I}$ has a certain combinatorial property. Naturally, the size of this coreset for $X$ depends on an upper bound on $|\mathcal{I}|$.

Second, since the choice of family $\mathcal{I}$ is oblivious to the data set, it suffices to design an efficient algorithm for $k$-CENTER coreset for any restricted data set. We observe that the efficiency bottleneck in Algorithm 2 is the repeated invocation of Algorithm $\mathcal{A}$ to construct a coreset, even though its input changes only a little between consecutive invocations. Hence, we design a dynamic algorithm, that maintains a $k$-CENTER coreset on the restricted data sets under point updates. Our algorithm may be viewed as a variant of Gonzalez's algorithm [Gon85], and we maintain it efficiently by a random projection idea that was used e.g. in [Ind03]. In particular, we "project" the data points onto several one-dimensional lines in $\mathbb{R}^d$, and we maintain an interval data structure (that is based on balanced trees) to dynamically maintain the result of our variant of Gonzalez's algorithm. We summarize the dynamic algorithm in the following lemma.

**Lemma A.6.** *There is a randomized dynamic algorithm with the following guarantees. The input is a dynamic set $X \subset \mathbb{R}_?^d$ of $j$-points, such that $X$ undergoes $q$ adaptive updates (point insertions and deletions) and the points ever added are fixed in advance (non-adaptively). The algorithm maintains in time $\tilde{O}\left( \frac{(j+k)^{j+k+1}}{j^j k^k} \cdot (j + k \log q)(d + k^2 \log q) \right)$ per update, a subset $Y \subseteq X$ of size $|Y| \leq O\left( \frac{(j+k)^{j+k+1}}{j^j k^{k-1}} \cdot \log d \right)$ such that with constant probability, $Y$ is an $O(k\sqrt{d \log q})$-coreset for $k$-CENTER on $X$ after every update.*

The proof of the lemma can be found in Section A.3, and here we proceed to the proof of Lemma A.4.

*Proof of Lemma A.4.* We plug in the dynamic algorithm in Lemma A.6 as $\mathcal{A}$ in Lemma A.5. Specifically, line 3 and 7 of Algorithm 2 are replaced by the corresponding query and update procedure. The detailed description can be found in Algorithm 3.

---

**Algorithm 3** Efficient importance score estimation

1: let $\mathcal{D}$ be the dynamic data structure defined in Algorithm 4, and call $\mathcal{D}$.INIT
2: $\forall x \in X$, insert $x$ to $\mathcal{D}$
3: $i \leftarrow 1$
4: **while** $X \neq \emptyset$ **do**
5:     $P \leftarrow \mathcal{D}$.GET-CORESET
6:     **for** $x \in P$ **do**
7:         $\sigma_x \leftarrow O(\alpha^z / i)$
8:     **end for**
9:     $\forall x \in P$, remove $x$ from $\mathcal{D}$
10:    $i \leftarrow i + 1$
11: **end while**
12: **return** $(\sigma_x : x \in X)$

---

Since $|X| = n$, and each point is inserted and deleted for exactly once, algorithm 2 needs $q = O(n)$ insertions and deletions of points. Moreover, the set of points ever added is just $X$ which is fixed. Thus, $\alpha$ is replaced by $O(k\sqrt{d \log n})$ and $T$ is replaced by $O\left(\frac{(j+k)^{j+k+1}}{j^j k^{k-1}} \cdot \log d\right)$. Therefore, for $(k, z)$-CLUSTERING, this computes $\sigma_x$ for $x \in X$ such that $\sigma_x \geq \sigma_x^\star$, and that

$$\sum_{x \in X} \sigma_x \leq \alpha^z \cdot T \cdot \log n = O\left(\frac{(j+k)^{j+k+1}}{j^j k^{k-z-1}} \cdot \sqrt{d^z \cdot \log^{z+2} n}\right).$$

The total running time is bounded by $\tilde{O}\left(\frac{(j+k)^{j+k+2}}{j^j k^{k-2}} \cdot nd\right)$ for implementing $O(n)$ updates. $\qquad\square$

## A.3 Proof of Lemma A.6: Dynamic $O(1)$-Coresets for $k$-Center Clustering

As mentioned, the high level idea is to identify a collection $\mathcal{I}$ of subsets of coordinates (so each $I \in \mathcal{I}$ satisfies $I \subseteq [d]$), construct an $\alpha$-coreset ($a$ will be determined is the later context) $Y_i$ for $k$-CENTER on the data set $X$ with coordinates *restricted* on each $I_i \in \mathcal{I}$, and then the union $\bigcup_i Y_i$ would be the overall $\alpha\sqrt{d}$-coreset for $k$-CENTER on $X$. The exact definition of restricted data set goes as follows.

**Definition A.3.** For a point $p \in \mathbb{R}^d_?$ and a subset $I \subseteq I_p$, define $p_{|I} \in \mathbb{R}^I$ in the obvious way, by selecting the coordinates $\{p_i\}_{i \in I}$. Define the *I-restricted data set* to be $X_{|I} := \{p_{|I} : p \in X, I \subseteq I_p\}$. Since each vector in $X_{|I}$ arises from a specific vector in $X$, a subset $Y \subseteq X_{|I}$ corresponds to a specific subset of $X$, and we shall denote this subset by $Y^{-1}$.

We observe that the metric space on the restricted data set becomes a usual metric space, i.e. it satisfies the triangle inequality, and can be realized as a point set in $\mathbb{R}^I$ which does not contain question marks. Therefore, this reduces our goal to constructing $k$-CENTER coresets for this usual data set. However, the size of the coreset yielded from this approach would depend on the size of the family $\mathcal{I}$. Hence, a key step is to identify a small set $\mathcal{I}$ such that the union of the coreset restricted on $\mathcal{I}$ is an accurate coreset. To this end, we consider the so-called $(j, k, d)$-family of coordinates as in Definition A.4. This family itself is purely combinatorial, but we will show in Lemma A.7 that such a family actually suffices for the accuracy of the coreset, and we show in Lemma A.8 the existence of a small family.

**Definition A.4.** A family of sets $\mathcal{I} \subset 2^{[d]}$ is called a $(j, k, d)$-family if for any $J, K \subset [d], J \cap K = \emptyset, |J| = j, |K| = k$, there exists an $I \in \mathcal{I}$ such that $I \cap J = \emptyset$ and $K \subset I$.

**Lemma A.7.** *Suppose $\mathcal{I}$ is a $(j, k, d)$-family Let $X \subseteq \mathbb{R}^d_?$ be a set of $j$-points, and for every $I \in \mathcal{I}$, let $Y_I$ be an $\alpha$-coreset for $k$-CENTER on $X_{|I}$. Then $\cup_{I \in \mathcal{I}} Y_I^{-1}$ is an $\alpha\sqrt{d}$-coreset for $k$-Center on $X$.*

*Proof.* It suffices to show that for any center set $C = \{c^1, \ldots, c^k\} \subseteq \mathbb{R}^d$ with $k$ points and $x \in X$, if $\mathrm{dist}(x, C) \geq r$ for some $r \geq 0$, then we can find a coreset point $y \in \cup_{I \in \mathcal{I}} Y_I^{-1}$ such that $\mathrm{dist}(y, C) \geq \frac{r}{\alpha\sqrt{d}}$.

For $i \in [k]$, let $t_i \in \arg\max_{t \in I_x} |x_t - c_t^i|$, i.e., $t_i$ is the index of coordinate that contributes the most in distance $\mathrm{dist}(x, c^i)$, so $|x_{t_i} - c_{t_i}^i| \geq \frac{r}{\sqrt{d}}$. Let $K$ be any $k$-subset such that $K \subseteq I_x$ and $\{t_1, \ldots, t_k\} \subseteq K$. Since $\mathcal{I}$ is a $(j, k, d)$-family and $|I_x| \geq d - j$, by definition, there exists an $I \subseteq \mathcal{I}$ such that $K \subseteq I \subseteq I_x$. We note that

$$\mathrm{dist}(x_{|I}, C_{|I}) = \mathrm{dist}_I(x, C) = \min_{i \in [k]} \mathrm{dist}_I(x, c^i) \geq \min_{i \in [k]} \mathrm{dist}_K(x, c^i) \geq \min_{i \in [k]} |x_{t_i} - c_{t_i}^i| \geq \frac{r}{\sqrt{d}}.$$

Since $I \subseteq I_x$, we know that $x_{|I} \in X_{|I}$. As $Y_I$ is an $\alpha$-coreset for $X_{|I}$, we know that there exists $y \in Y_I^{-1}$ such that

$$\mathrm{dist}(y, C) \geq \mathrm{dist}_I(y, C) = \mathrm{dist}(y_{|I}, C_{|I}) \geq \frac{\mathrm{dist}(x_{|I}, C_{|I})}{\alpha} \geq \frac{r}{\alpha\sqrt{d}}.$$

$\qquad\square$

Next, we show the existence of a small $(j, k, d)$-family. We remark that this combinatorial structure has been employed in designing fault-tolerant data structures and algorithms (cf. [DK11, DGR21,

KP21]). Similar bounds were obtained in their different contexts and languages, and here we provide a proof for completeness.

**Lemma A.8.** *There is a $(j, k, d)$-family $\mathcal{I}$ of size $O\left(\frac{(j+k)^{j+k+1}}{j^j k^k} \log d\right)$. Moreover, there is a randomized algorithm that constructs $\mathcal{I}$ in time $O(d \cdot |\mathcal{I}|)$ with probability at least $1 - \frac{1}{d^{j+k}}$.*

*Proof.* Set $t = \frac{(j+k)^{j+k+1}}{j^j k^k} \cdot 2 \log d$. We add $t$ random sets into $\mathcal{I}$ where each random set is generated by independently including each element of $[d]$ with probability $\frac{k}{j+k}$. For a set $J \subseteq [d], |J| = j$ and a set $K \subseteq [d], |K| = k$ such that $J \cap K = \emptyset$, the probability that a random set generated in the above way contains $K$ but avoids $J$, is

$$\left(\frac{j}{j+k}\right)^j \cdot \left(\frac{k}{j+k}\right)^k.$$

Since there are at most $d^{j+k}$ tuples of such $J$ and $K$, by union bound and the choice of $t$, the probability that $\mathcal{I}$ is a $(j, k, d)$-family is at least

$$1 - d^{j+k} \left(1 - (\frac{j}{j+k})^j \cdot (\frac{k}{j+k})^k\right)^t \geq 1 - \frac{1}{d^{j+k}}$$

$\square$

**Gonzalez's algorithm yields $k$-CENTER coreset for restricted data set.** Finally, the $k$-CENTER coreset for the restricted data set on each $I \in \mathcal{I}$ would be constructed using an approximate version of Gonzalez's algorithm [Gon85]. We note that while Gonzalez's algorithm was originally designed as an approximation algorithm for $k$-CENTER, the approximate solution actually serves as a good coreset for $k$-CENTER (see Lemma A.9). The assumption that the input forms a metric space is crucial in Lemma A.9, and this is guaranteed since we run this variant of Gonzalez only on a restricted data set which satisfies the triangle inequality.

**Lemma A.9** (Approximate Gonzalez). *Let $(M, d)$ be a metric space. Let $A \subset M$ be a set of $n$ points and consider the following variant of Gonzalez's greedy algorithm. Set $B = \{b_0\}$ for an arbitrary $b_0 \in A$. Repeat for $k$ times, where each time we add a $c$-approximation of $B$'s furthest point into $B$. Precisely, add $b_i \in A$ such that $c \cdot \mathrm{dist}(b_i, B) \geq \max_{a \in A} \mathrm{dist}(a, B)$ into $B$. Then $B$ is a $(1 + 2c)$-coreset for $k$-CENTER on $A$.*

*Proof.* Fix a center set $C = \{c_1, \ldots, c_k\}$ with $k$ points and let $r := \max_{b \in B} \mathrm{dist}(b, C)$. Then we have $\bigcup_{i=1}^k \mathrm{Ball}(c_i, r)$ covers $B$ where $\mathrm{Ball}(x, r) = \{y : \mathrm{dist}(x, y) \leq r\}$ is the ball centered at $x$ with radius $r$. It suffices to prove that $A \subseteq \bigcup_{i=1}^k \mathrm{Ball}(c_i, (2c+1)r)$.

Since $k$ balls $B(c_1, r), \cdots, B(c_k, r)$ cover $B$ and $|B| = k + 1$, by pigeonhole principle, there exists $b_i, b_j \in B, i < j$ that are contained in a same ball $B(c_i, r)$. W.l.o.g., we assume $b_i, b_j \in B(c_1, r)$. Now fix $a \in A \setminus B$, since $a$ has never been added into $B$, we have

$$\begin{aligned}
\mathrm{dist}(a, B) &\leq \mathrm{dist}(a, \{b_1, \ldots, b_{j-1}\}) \\
&\leq c \cdot \mathrm{dist}(b_j, \{b_1, \ldots, b_{j-1}\}) \\
&\leq c \cdot \mathrm{dist}(b_i, b_j) \\
&\leq c \cdot (\mathrm{dist}(b_i, c_1) + \mathrm{dist}(b_j, c_1)) \\
&\leq 2cr.
\end{aligned}$$

Thus $A \subseteq \bigcup_{i=1}^{k+1} \mathrm{Ball}(b_i, 2cr) \subseteq \bigcup_{i=1}^k \mathrm{Ball}(c_i, (2c+1)r)$. $\square$

**Dynamic implementation of Gonzalez's algorithm.** To make this $k$-CENTER coreset construction dynamic, we adapt the random projection technique to Gonzalez's algorithm, so that it suffices to dynamically execute Gonzalez's algorithm on a set of one-dimensional lines in $\mathbb{R}^d$.

**Random projection.** We call a sample from the $d$-dimensional standard normal distribution $N(0, I_d)$ a $d$-dimensional *random vector* for simplicity. To implement (the variant of) Gonzalez's algorithm as in Lemma A.9 in the dynamic setting, we project the point set to several random vectors and use one dimensional data structure to construct $k$-CENTER coreset in each of the one dimensional projected data set.

Note that the key step in Gonzalez's algorithm is the furthest neighbor search, and we would show that our projection method eventually yields an $O(k\sqrt{\log n})$-approximation of the furthest neighbor with high probability. The following two facts about normal distribution are crucial in our argument, and Lemma A.12 is our main technical lemma.

**Fact A.10.** *Let $u \in \mathbb{R}^d$ and let $v \sim N(0, I_d)$ be a random vector, then $\langle u, v/|u| \rangle \sim N(0,1)$.*

**Fact A.11.** *Let $Z \sim N(0,1)$, then there exists some universal constant $c > 0$ such that $P[|Z| \leq \frac{1}{k}] \leq \frac{c}{k}$, and $P[|Z| \geq t] \leq e^{-c \cdot t^2}$ for any $t > 0$.*

**Lemma A.12.** *Let $X \subset \mathbb{R}^d, |X| = n, \delta > 0$ and integer $k \geq 1$. Let $\mathcal{V}$ be a collection of $t = O(k \log n + \log \delta^{-1})$ random vectors in $\mathbb{R}^d$. Then with probability $1 - \delta$, for every $C \subseteq X, |C| \leq k$ and every $x \in X$, there exists a vector $v \in \mathcal{V}$ such that (i) $|x \cdot v - c \cdot v| \geq \Omega(\frac{1}{k}) \cdot \|c - x\|_2$ for every $c \in C$ and (ii) $|a \cdot v - b \cdot v| \leq O(\sqrt{\log n}) \cdot \|a - b\|_2$ for every $a, b \in X$.*

*Proof.* Fix a subset $C \subseteq X, |C| \leq k$, a point $x$ and a random vector $v$. For every $c \in C$, since $(c - x) \cdot v/\|c - x\|_2 \sim N(0,1)$, by Fact A.11, the probability that $|c \cdot v - x \cdot v| \geq \Omega(\frac{1}{k}) \cdot \|c - x\|_2$ is at least $1 - \frac{1}{4k}$. For every $a, b \in X$, since $(a - b) \cdot v/\|a - b\|_2 \sim N(0,1)$, by Fact A.11, the probability that $|a \cdot v - b \cdot v| \leq \sqrt{\log n}\|a - b\|_2$ is at most $\frac{1}{4n^2}$.

Since there are $k$ choices of $c \in C$ and at most $n^2$ choices of $a, b \in X$, by union bound, with probability at least $1 - k \cdot \frac{1}{4k} - n^2 \cdot \frac{1}{4n^2} = \frac{1}{2}$, the following two events hold, (i) $|x \cdot v - c \cdot v| \geq \Omega(\frac{1}{k}) \cdot \|c - x\|_2$ for every $c \in C$ and (ii) $|a \cdot v - b \cdot v| \leq O(\sqrt{\log n}) \cdot \|a - b\|_2$ for every $a, b \in X$.

Now since $\mathcal{V}$ contains $t$ random vectors, the probability that there exists one vector $v \in \mathcal{V}$ that satisfies (i) and (ii) is at least $1 - \frac{1}{2^t}$.

Finally, by union bound, since there are at most $(n + 1)^{k+1}$ choices of $C \subseteq X, |C| = k$ and $x \in X$, the probability such that for every $C$ and $x$, there exists $v \in \mathcal{V}$ such that (i) and (ii) happen is at least $1 - \frac{(n+1)^{k+1}}{2^t} \geq 1 - \delta$. $\qquad\square$

In the next lemma, we present a dynamic algorithm that combines the random projection idea with a one-dimensional data structure. This combining with the $(j, k, d)$-family idea would immediately imply Lemma A.6.

**Lemma A.13.** *There is a dynamic algorithm that for every $P \subseteq \mathbb{R}^m$ subject to at most $q$ adaptive point insertions and deletions where the set of points ever added is fixed in advance, and every $\delta > 0$, maintains set $Q \subseteq P$ with $|Q| \leq k + 1$ such that with probability at least $1 - \delta$, $Q$ is an $O(k\sqrt{\log q})$-coreset for $k$-CENTER on $P$ after every update, in time $O\big((k^2 \log q + m)(k \log q + \log \delta^{-1})\big)$ per update.*

*Proof of Lemma A.6.* We present our dynamic algorithm in Algorithm 4.

**Analysis.** Since we pick $\delta = \Theta\left(\frac{1}{|\mathcal{I}|}\right)$ for all $\mathcal{D}_I$'s, with constant probability all data structures $\mathcal{D}_I$'s succeed simultaneously. The running time follows immediately from Lemma A.8 and Lemma A.13. The coreset accuracy follows from Lemma A.7 and Lemma A.13 (noting that we need to suffer a $\sqrt{d}$ factor because of Lemma A.7). $\qquad\square$

*Proof of Lemma A.13.* We assume there is a data structure $\mathcal{T}$ that maintains a set of real numbers and supports the following operations, all running in $O(\log n)$ time where $n$ is the number of elements currently present in the structure.

- REMOVE($x$): Remove an element $x$ from the structure.

- ADD($x$): Add an element $x$ to the structure.

---

**Algorithm 4** Dynamic $k$-CENTER coreset with missing values

---

1: **procedure** INIT
2:   let $\mathcal{I}$ be a $(j, k, d)$-family generated by sampling, as in Lemma A.8
$$\triangleright |\mathcal{I}| = O\left(\frac{(j+k)^{j+k+1}}{j^j k^k} \log d\right)$$
3:   $\forall I \in \mathcal{I}$, initialize data structure $\mathcal{D}_I$ using Algorithm 5 (Lemma A.13) with failure probability
   $\delta := \Theta\left(\frac{1}{|\mathcal{I}|}\right)$, and initialize $Y_I = \emptyset$
4: **end procedure**
5: **procedure** UPDATE($x \in \mathbb{R}_?^d$)
6:   **for** $I \in \mathcal{I}$ **do**
7:     $\mathcal{D}_I$.UPDATE($x_{|I}$)
8:     $Y_I \leftarrow \mathcal{D}_I$.GET-CORESET($k$)
9:   **end for**                                      $\triangleright$ we use UPDATE and Get-Coreset in Algorithm 5
10: **end procedure**
11: **procedure** GET-CORESET
12:   return $\bigcup_{I \in \mathcal{I}} Y_I^{-1}$                        $\triangleright$ as in Lemma A.7
13: **end procedure**

---

- UPPERBOUND($x$): Return the largest element that is at most $x$.

- LOWERBOUND($x$): Return the smallest element that is at least $x$.

Note that such $\mathcal{T}$ may be implemented by using a standard balanced binary tree.

**Furthest point query.** We also need FURTHEST($C$) query, where $C \subset \mathbb{R}$ and it asks for an element $x$ that has the largest distance to $C$ (and it should return an arbitrary element if $C = \emptyset$). This FURTHEST($C$) can be implemented by using $O(|C|)$ many UPPERBOUND and LOWERBOUND operations, which then takes $O(|C| \log n)$ time in total. To see this, assume $C = \{c_1, \ldots, c_k\}$ where $c_1 \leq \ldots \leq c_k$ then the clusters partitoned by $C$ is $(-\infty, \frac{1}{2}(c_1 + c_2)], (\frac{1}{2}(c_1 + c_2), \frac{1}{2}(c_2 + c_3)], \cdots, (\frac{1}{2}(c_{k+1} + c_k), +\infty)$ and we can find the potential furthest points in each cluster by querying the following,

$$\text{UPPERBOUND}(-\infty), \text{LOWERBOUND}\left(\frac{1}{2}(c_1 + c_2)\right),$$

$$\text{UPPERBOUND}\left(\frac{1}{2}(c_1 + c_2)\right), \text{LOWERBOUND}\left(\frac{1}{2}(c_2 + c_3)\right)$$

$$\cdots$$

$$\text{UPPERBOUND}\left(\frac{1}{2}(c_{k+1} + c_k)\right), \text{LOWERBOUND}(+\infty)$$

and the furthest point to $C$ among the above $2k = O(|C|)$ many points is what we seek for.

The dynamic algorithm is presented in Algorithm 5. The algorithm samples a set of independent random vectors $\mathcal{V}$ (in a data oblivious way), then creates an above-mentioned interval structure $\mathcal{T}_v$ for each $v \in \mathcal{V}$. When we insert/delete a point $x$, the update is performed on every $\mathcal{T}_v$ with the projection $\langle x, v \rangle$. The coreset for the current data set $P$ can be computed on the fly by simulating the Gonzalez's algorithm. In particular, this is where the Furthest query is used, and we find an approximate furthest point in $P$ by taking the furthest point in each $\mathcal{T}_v$, and select the one that is the relative furthest in $P$.

**Analysis.** Let $A$ be the set of points ever added, so $|A| \leq q$. Recall that $A$ is fixed in advance. By applying Lemma A.12 in $A$, we know that with probability $1 - \delta$, the following event $\mathcal{E}$ happens. For every $C \subseteq A, |C| \leq k$, every $x \in A$, there exists $v \in \mathcal{V}$, such that

(i) $|\langle c - x, v \rangle| \geq \Omega(\frac{1}{k}) \cdot \|x - c\|_2$ for every $c \in C$, and

(ii) $|\langle a - b, v \rangle| \leq O(\sqrt{\log q}) \cdot \|a - b\|_2$ for every $a, b \in A$.

---

**Algorithm 5** Dynamic Gonzalez's algorithm

---

1: **procedure** INIT                                      ▷ initialize an empty structure
2:      $l \leftarrow O(k \log q + \log \delta^{-1})$, and draw $l$ independent random vectors in $\mathbb{R}^m$, denotes as $\mathcal{V}$
3:      initialize $\mathcal{T}_v$ for each $v \in \mathcal{V}$
4: **end procedure**
5: **procedure** UPDATE($x$)
6:      insert/delete $\langle x, v \rangle$ for each $v \in \mathcal{V}$
7: **end procedure**
8: **procedure** GET-CORESET($k$)
9:      $Q \leftarrow \emptyset$
10:     **for** $i = 1, \ldots, k+1$ **do**
11:        for $v \in \mathcal{V}$, let $x_v \in P$ satisfy $\langle x_v, v \rangle = \mathcal{T}_v.\text{FURTHEST}(\langle Q, v \rangle)$
                                        ▷ where $\langle Q, v \rangle := \{\langle x, v \rangle : x \in Q\}$
12:        $v^\star \leftarrow \arg \max_{v \in \mathcal{V}} \text{dist}(x_v, Q)$
13:        $Q \leftarrow Q \cup \{x_{v^\star}\}$
14:     **end for**
15:     **return** $Q$
16: **end procedure**

---

Now condition on $\mathcal{E}$. Suppose the current point set is $P$. Suppose we run the GET-CORESET subroutine and we query $\mathcal{T}_v.\text{FURTHEST}(\langle Q, v \rangle)$ for some $v$ and $Q$. Suppose $x \in P \subseteq A$ is the current furthest point to $Q$. Because of $\mathcal{E}$, there exists a vector $v \in \mathcal{V}$ such that (i) and (ii) hold. By (i), we have that $\text{dist}(\langle x, v \rangle, \langle Q, v \rangle) \geq \Omega(\frac{1}{k}) \cdot \text{dist}(x, Q)$. By (ii), we know that for any $p \in P$ and $c \in Q$, $|\langle p - c, v \rangle| \leq O(\sqrt{\log q})\|p - c\|_2$, so $\text{dist}(\langle p, v \rangle, \langle Q, v \rangle) \leq O(\sqrt{\log q}) \cdot \text{dist}(p, Q)$. So if $\mathcal{T}_v.\text{FURTHEST}(\langle Q, v \rangle)$ returns an answer $\langle p, v \rangle$, we know that

$$\text{dist}(p, Q) \geq \frac{\text{dist}(\langle p, v \rangle, \langle Q, v \rangle)}{O(\sqrt{\log q})} \geq \frac{\text{dist}(\langle x, v \rangle, \langle Q, v \rangle)}{O(\sqrt{\log q})} \geq \Omega\left(\frac{1}{k\sqrt{\log q}}\right) \cdot \text{dist}(x, Q).$$

Thus, $p$ is an $O(k\sqrt{\log q})$-approximation of the furthest point to $Q$. This combining with Lemma A.9. implies the error bound.

**Running time.** For the running time, we note that for each update of $P$, we need to update $\mathcal{T}_v$ for each $v \in \mathcal{V}$ accordingly. Thus we need to pay $O(lm)$ time (recalling that $l = O(k \log q + \log \delta^{-1})$ was defined in Algorithm 5) to compute all the inner products and $O(l \log q)$ time to update all $\mathcal{T}_v$'s. The main loop in GET-CORESET requires $O(kl)$ many FURTHEST$(\cdot)$ queries and this runs in $O(k^2 l \log q)$ time in total. In conclusion, the running time of each update (and maintaining coreset) is bounded by

$$O\left((k^2 \log q + m) \cdot l\right) = O\left((k^2 \log q + m)(k \log q + \log \delta^{-1})\right).$$

$\square$

# B   Lower Bound

We prove the following lower bound to assert the necessity of the exponential dependence on $\min(j, k)$ in our coreset construction Theorem 3.1.

**Theorem B.1** (Restatement of Theorem 1.2)**.** *Consider the $k$-MEANS with missing values problem in $\mathbb{R}^d_?$ where each point can have at most $j$ missing coordinates. Assume there is an algorithm that constructs an $\epsilon$-coreset of size $f(j, k) \cdot \text{poly}(\epsilon^{-1} d \log n)$, then $f(j, k)$ can not be as small as $2^{o(\min(j,k))}$.*

*Proof.* Consider the following $n$ points instance with $j = k = \Theta(\log n)$, and $d = 2j$. For a subset $I$ of $[d]$, we define a data point $p(I)$ such that $p(I)_i = 1$ if $i \in I$ and $p(I)_i = ?$ otherwise. Then we let the data set $P = \{p(I) | I \subseteq [d], |I| = j\}$. We remark that we can make $|P| = \binom{d}{j} = n$ by choosing a proper $j = \Theta(\log n)$.

We prove that any $1/2$-coreset of $P$ should contain every point in $P$. Let $D$ be such a coreset and assume $p(I) \notin D$, we choose the following $k = j$ centers. For every $i \in I$, we define a center $c^i \in \mathbb{R}^d$ such that the $i$-th coordinate of $c^i$ is $0$ and the other coordinates of $c^i$ are $1$. We observe that, for any $i \in I$, $\mathrm{dist}(p(I), c^i) = 1$. Meanwhile for any other $p(I') \neq p(I)$, there must be a $i' \in I \setminus I'$ since $|I| = |I'|$, thus $\mathrm{dist}(p(I'), c^{i'}) = 0$. This should imply that the cost on coreset is $0$ while the cost on $P$ is $1$ which makes a contradiction.

Since $j = k = \Theta(\log n)$, $d = 2j$, we have $2^{o(\min(j,k))} \cdot \mathrm{poly}(d \log n) = o(n)$. Thus $f(j,k)$ can not be as small as $2^{o(\min(j,k))}$. □