# OpenReview forum: "Coresets for Clustering with Missing Values"
_NeurIPS.cc/2021/Conference — NeurIPS 2021 Spotlight_

### Official Review · Reviewer_Lwwe · 2021-07-05

**Rating:** 8
**Confidence:** 4

**Summary:**

The paper describes the first algorithms for coresets for k-means clustering with missing data coordinates for some points in d-dimensions, with provably guarantees.  The algorithm runs in linear time O(nd) times factor depending only on j (most possible missing values per coordinate) and k (number of allowed clusters).  A SODA 2021 paper provides a PTAS for this problem, but it has worse dependence on j, k, epsilon and has a base cost of O(n^2 d), so quadratic in n.  This paper demonstrates their approach is implementable, and practical.


**Limitations And Societal Impact:**

I was glad to see the paper had (short) sections addressing these points, even though as a mostly core/theory paper the real impacts are distant.  They could have also mentioned issues with imputing missing data, but it is forgivable since again the impact of this paper on actually societal issues is quite hard to extrapolate.

**Main Review:**

Coresets for k-means clustering (and related problems) is a heavily studied area, and some improvements in paper seems incremental and not so consequential.  I feel this paper is *not* of that sort.  Missing values in data is a real and serious issue, and this paper basically provides the first scalable solution.  The main innovation is a provably accurate way to construct the coreset that then integrates (and relies on) natural and standard algorithms (Gonzalez & Lloyds, etc).

Moreover, I found the writing of the paper good.  It does not develop any especially deep or important techniques (although the dynamic k-center result is something), but it expertly borrows and builds on several choice advances from the literature.  Sometimes non-standard approaches were needed, and the paper does a good job of pointing out what was standard and what needed extra care and development.  After highlighting the main theoretical developments, it leaves space to carefully describe the experiments which do a good job of demonstrating the algorithms empirical advantages.


I do have one main complaint/suggestion.  The overall algorithm is not totally clear.  Section 1.2 does an ok job of overviewing it, but that section also provides intuition for various choices and where simple techniques fail.  I suggest the paper adds a short section just outlining the main algorithm -- there should be space.


**Time Spent Reviewing:**

2

---

> ### Author Response · Authors · 2021-08-09
> **Response to the Fourth Reviewer**
>
> * We will add an explicit and self-contained description of the algorithm in the revision.

---

> > ### Comment · Reviewer_Lwwe · 2021-08-13
> > **comment response**
> >
> > Seems we are in agreement this paper should be accepted.  I see no major concerns, and agreement the paper is interesting.

---

### Official Review · Reviewer_wxoY · 2021-07-12

**Rating:** 8
**Confidence:** 4

**Summary:**

The paper provides the first coreset for clustering tasks, where each of the input d-dimensional points might contain up to j missing values. The coreset construction time is near-linear. The main technical result of the paper is a reduction from the above problem, to the problem of computing a dynamic coreset for the k-center problem with missing values. The authors utilize the sensitivity sampling framework for this task.
Using this coreset, the authors improve the running time of an existing PTAS for the problem, from quadratic, to near-linear time. The authors provide experimental results which demonstrate the effectiveness of the coreset in practice, as compared to a uniform random sample.

**Limitations And Societal Impact:**

This has been addressed.

**Main Review:**

Pros:
- In overall I really like this paper. The reduction to the dynamic k-center problem as well as the solution for this problem via the restriction of the input set into multiple sets with not missing values is very interesting.
- The paper is mostly well written.
- The paper presents significant improvement of the state of the art for this problem.
- The provided coreset is original, and seems very useful for real-world problems, as shown in the experimental results section.

Detailed comments and suggestions:
- I did not find the exact definition of \tilde{O}, which is very important.
- There is a typo in the legend of Fig. 4(b).
- The comparison to the previous work of MF19 is not very accurate. MF19 handles any input set of lines, which are not necessarily axis-parallel. The proposed work handles j-dimensional subspaces (for j >=1), but which are axis parallel.
- It was difficult to sum-up the final coreset construction algorithm, as it is spread across the whole paper. A stand-alone pseudo-code would be very helpful.
- In some claims the dependency on some constants, e.g., the probability of failure \delta, is explicitly stated, while in others it is not. I would suggest explicitly adding the dependency on all parameters / constants in all the claims.
- The standard deviation in the experiments is reported only for the Russian housing dataset. Is there a reason why this is not reported for the other datasets?
- Missing a conclusions and future work section.

**Time Spent Reviewing:**

5

---

> ### Author Response · Authors · 2021-08-09
> **Response to the Third Reviewer**
>
>
> * For the deviation in the experiments, we provided variance only for the RussianHouse experiment in the submission due to page limit. In fact, our coreset achieves a smaller variance than uniform sampling in every experiment. We will add the other plots in the revision.
>
> * We will try to discuss the future work in the revision. For instance, this could include our insights about the lower bounds and other possibilities of tradeoffs of parameters.
>
> For the minor comments:
>
> * We will add an exact definition for $\tilde{O}$ in the revision.
> * We will fix the typo in the legend of Fig 4(b) in the revision.
> * We will add a more detailed and accurate discussion about MF19 in the revision.
> * We will add an explicit and self-contained description of the algorithm in the revision.
> * We will make the dependency on all parameters clear in the statements of claims in the revision.

---

### Official Review · Reviewer_DJy9 · 2021-07-12

**Rating:** 8
**Confidence:** 3

**Summary:**

The paper suggests a coreset for clustering points in $\mathbb{R}^d$ that have multiple missing values (coordinates) including K-means and K-median.

**Limitations And Societal Impact:**

1. The coreset size is exponential in k and j, where k is the number of clusters and j is the number of missing values. This is a large bound when both j and k are large values.
----> My question is how hard the problem really is? what is the lower bound for the coreset?

2. Please address the novelty question above.

3. Experimental results: please add variance to the graphs.

4. Missing conclusion and future work section.





**Main Review:**

The paper gives the first coreset handling clustering points in $\mathbb{R}^d$ that have multiple missing values.
The coreset construction is based on the importance sampling framework introduced by Feldman and Langberg. The main challenge in this coreset with missing values is that distances do not satisfy the triangle inequality, hence importance scores cannot be easily computed. To overcome this hurdle, the authors used a method introduced by Varadarajan and Xiao for projective clustering.

I enjoyed reading the paper, the paper is well written, the results are interesting (also very strong), and the authors evaluate the performance of their results on real and synthetic datasets.


comments:

1) Very important: please provide a detailed paragraph about the novelty of your work.
It seems that the paper relies on other papers for suggesting the coreset, while I believe that there is something very strong hidden in the approach --> please elaborate more on this point, since it is very important.

2) Remove the additional n in "Running" in Figure 4 (b).

**Time Spent Reviewing:**

5 hours

---

> ### Author Response · Authors · 2021-08-09
> **Response to the Second Reviewer**
>
> * We summarize our novelty in the following paragraph, which we will add to our paper. These points are correctly mentioned in the other reviews.
>
> Our coreset construction builds upon the sensitivity-sampling method, e.g., [Feldman, Langberg 2011]. However, a central technical challenge is that the standard method to compute the sensitivity scores breaks, because distances between points with missing values do not satisfy the triangle inequality. We overcome this using another known method, of [Varadarajan, Xiao 2012], that requires a coreset for k-center. Our main innovation is a near-linear time algorithm that computes an O(1)-approximate k-center coreset for points with missing values. To this end, we need the following key steps, which constitute our main technical contribution.
> We reduce the k-center coreset construction with missing values, to the construction of traditional k-center coresets (i.e., without missing values) on a series of instances. These instances are built by restricting data points with missing values to a carefully-chosen collection of subspaces. The guarantee needed from this collection is a certain combinatorial structure, and we indeed prove it exists.
> The method of Varadarajan and Xiao executes the k-center coreset algorithm many times, and overall takes quadratic time. To improve the running time, we design an efficient dynamic algorithm for the well-known Gonzales’ algorithm (which computes an O(1)-approximate k-center coreset). The main idea in this dynamic algorithm is to project the data points onto (data-oblivious) random 1D lines, and build on each line a dynamic data structure that supports furthest-neighbor queries (in 1D).
> Finally, we implemented our algorithm and the experiments indicate that our algorithm is efficient and accurate enough to be potentially applicable in practice.
>
> * For the lower bound, we recently obtained one that justifies our exponential dependence on $min(j,k)$, and we will add this to the paper. The proof is short but a bit tricky, and shows inputs with $d,j,k = \Theta(log n)$, for which every 0.5-coreset must have size $\Omega(n) \geq \Omega(exp( min(j,k) ))$.
>
> * We provided variance only for the RussianHouse experiment in the submission due to the page limit. In fact, our coreset achieves a smaller variance than uniform sampling in every experiment. We will add the other plots in the revision.
>
> * We will try to discuss the future work in the revision. For instance, this could include our insights about the lower bounds and other possibilities of tradeoffs of parameters.
>
> * We will address other minor comments in the revision.

---

> > ### Comment · Reviewer_DJy9 · 2021-08-12
> > **Thanks for the clarifications**
> >
> >
> > I read the author's response and the other reviews. Thanks for the detailed answers to my concerns.
> >
> > The lower bound and the detailed novelty of course just strengthing my support to increase my score.

---

### Official Review · Reviewer_7KH2 · 2021-07-22

**Rating:** 9
**Confidence:** 4

**Summary:**

This work considers a setup when points might have coordinates that are missing, and the distance between them is defined only by using the filled-in coordinates that they have in common. Note that this means that the distance function is not even a metric, and indeed has very few nice properties that we can depend upon.

**Limitations And Societal Impact:**

No significant societal impact. Limitations include no known lower bound, especially figuring out whether the exponential dependence on j & k is needed. Other limitations include the experimental results, as mentioned above.

**Main Review:**

This is the first work to give coresets for multiple missing coordinates, and it improves the coreset size upon existing results even for the one missing coordinate case.

In most papers regarding coresets, there are two meta approaches -- either via aggregation using a fine mesh, or via using importance sampling, especially using the framework developed by Langberg et al. This work uses an existing framework to reduce the importance-score computation to the computation of k-center coresets,and being able to compute these k-center coresets for missing coordinates is the main technical contribution. At its heart is a nice implementation of an existing algorithm that makes it more efficient. I also found the restriction into the collection of subsets to be a very interesting construction.

Overall the work seems quite novel and is nicely written. My only slight crib is about the experiments. While uniform sampling is the most obvious baseline to compare with in this setting, perhaps it would be useful to see some other baselines that might not have a theoretical guarantee (at least in the worst-case setting). For instance, filling in the missing values using a suitable heuristic, and then using the standard k-means coreset -- how will that perform in practice.


**Time Spent Reviewing:**

2

---

> ### Author Response · Authors · 2021-08-09
> **Response to the First Reviewer**
>
> * Thanks for the suggestion about adding a new baseline, and we will indeed add to the paper a comparison to this suggested baseline. In particular, we will fill missing values using a standard imputation method and apply the well-known importance-sampling coreset (e.g., [Feldman and Langberg ‘11]). Our preliminary experiments indicate that the performance of this new baseline is better than uniform sampling but fall behind our coreset.
>
> * We also have some recent progress regarding the lower bound. In particular, we managed to prove a lower bound that justifies our exponential dependence on $min(j,k)$, and we will add this to the paper. The proof is short but a bit tricky, and shows inputs with $d,j,k = \Theta(log n)$, for which every 0.5-coreset must have size $\Omega(n)\geq \Omega(exp( min(j,k) ))$.

---

### Decision · Program_Chairs · 2021-09-27

**Decision:**

Accept (Spotlight)

**Comment:**

The paper provides core set construction for clustering such as k means with multiple missing values. All the reviews are highly positive and support acceptance. The authors should carefully address all the comments that the reviewers have brought up including adding new baselines to their experimental evaluation.